# Exploring determinants of formation of cognitive anchors from altruistic messages: A fuzzy DEMATEL approach

Chi-Horng Liao[1,2,3]*, Chu-Chia Hsu[2]

**1** Department of Communication Studies, Tzu Chi University, Hualien City, Taiwan, **2** Bachelor Program in Digital Media and Technology, Tzu Chi University, Hualien City, Taiwan, **3** Media Production and Education Center, Tzu Chi University, Hualien City, Taiwan

* lchjerry@mail.tcu.edu.tw

## Abstract

Altruistic communication by non-profit organizations plays a crucial role in shaping individuals' perceptions and beliefs about altruism. One of the indicators of effective communication is the anchoring of the messages. Therefore, understanding the underlying determinants of anchoring in altruistic communication is essential. Despite the importance of anchoring in the communication of altruism, extant research has not done much to examine the determinants of anchoring in altruistic communication. This paper investigates the determinants of anchoring in non-profit organizations' altruistic communication through the lens of the dual process theory. It applies the Fuzzy Decision Making Trial and Evaluation Laboratory (F-DEMATEL) method to analyze the causal and effect factors. Data were gathered from 12 social communication experts based in Taiwan. Out of the 12 proposed determinants, three factors, namely consistency, cultural consideration, and emotional anchoring, were established as significant causal factors. Consistency had causal effects on five other factors, namely, the use of metaphors, the use of antinomies, thematic anchoring, understanding the cognitive ability of the audience, and crafting engaging information. Cultural consideration had causal effects on feedback, naming, use of antinomies, thematic anchoring, emotional anchoring, and repetition. Emotional anchoring had causal effects on thematic anchoring, use of antinomies, use of metaphors, consistency, naming, feedback, understanding the cognitive ability of the audience, and repetition. On the other hand, feedback, naming, and use of antinomies were established as significant effect factors. The study's findings offer crucial contributions to the social communication literature and provide important insights for social communication practitioners.

## 1. Introduction

In today's rapidly evolving communication and philanthropic landscape, non-profit organizations play an important role in driving social change and addressing social issues [1]. Public engagement and public perceptions are also crucial for the success of non-profit organizations [2]. For organizations to achieve public engagement and positive public perceptions, effective

**Data Availability Statement:** All relevant data are within the manuscript and its Supporting information files.

**Funding:** Chi-Horng Liao received funding from Tzu Chi Cultural and Communication Foundation,

and Yin Shun and Cheng Yen Education Foundation. The funders had no role in study design, data collection and analysis, decision to publish, or preparation of the manuscript.

**Competing interests:** The authors have declared that no competing interests exist.

communication is important. Effective communication enables organizations to raise awareness, inspire action, and cultivate support for their altruistic causes [3, 4]. Anchoring is a cognitive phenomenon associated with an audience's receptivity to information [5]. By priming individuals with initial information, anchoring shapes perceptions and receptivity toward subsequent information [6, 7]. Organizations use different means of communication to relay information about their work to stakeholders. One of the widely used means of communication is television [1, 8]. Television's unique attributes–visual narratives, emotional storytelling, and brand reach, interact with anchoring phenomena in distinctive ways [9, 10]. Understanding the factors leading to anchoring effects within television can illuminate how audiences perceive non-profit organizations' altruistic communications.

Prior research suggests that recipients retain messages if they stimulate cognitive processing and if they appeal to the emotions of the audience [11–14]. Individuals develop a personal connection with information that appeals to their emotions because the information is congruent with their state and is relatable to them, enhancing the memorability of the information [15]. On the other hand, cognitive processing of information entails active thinking and reasoning over information, which leads to better retainment of the information [11]. One theory that explains the processing of emotional and cognitive cues is the dual process theory. The theory explains how people process information via the central and peripheral processing paths [16]. The central path involves quick, intuitive, and emotional processing, whereas the peripheral path involves slower, deliberate, and analytical processing [17]. Although central and peripheral processing is critical in communication effectiveness, research is yet to establish whether the audience can anchor messages that appeal to emotions and cognitions. Given how important anchoring is in ensuring the effective processing of messages and how emotional and cognitive appeal shapes communication effectiveness, it is crucial to understand whether and how emotional and cognitive factors lead to anchoring of messages.

Prior studies have primarily examined the determinants of anchoring in communication using the experimental design [6, 9, 18]. Whereas the approaches used by previous studies have helped generate more knowledge about anchoring, they cannot reveal the dynamics of the complex relationships among factors, as is the case with F-DEMATEL [8]. In Addition, not all variables can be easily manipulated in research and practice for ethical and practical reasons. Furthermore, communication is a goal-oriented function that involves decision making support. Multicriteria decision methods such as F-DEMATEL identify key factors and assess the factors' importance and interrelationships in the communication context, which could help researchers and practitioners understand which factors hold higher degrees of importance [19]. As such, by applying the F-DEMATEL method, this study makes a methodological contribution to the communication literature by examining the determinants of anchoring using a relatively underutilized method.

The primary objective of this study is to examine the determinants of anchoring altruistic communication within the lens of the dual process theory. The study makes two key contributions to the social communication literature. First, applying the dual process theory demonstrates how peripheral and central factors can lead to anchoring altruistic messages. Although prior research has examined the determinants of anchoring in communication, little has been done to examine how anchoring can be achieved via the dual paths of cognition and emotion, which are known to affect communication effectiveness [11, 15]. Secondly, by applying the F-DEMATEL method, this study applies a relatively underutilized approach. By applying this method, the study demonstrates how the factors affect anchoring and the interrelationships among the factors. The study clarifies the complex interrelationships among the factors examined in the altruistic communication process. The practical significance of the study lies in the fact that it shows how non-governmental organizations and media practitioners can

communicate to their target audiences effectively to ensure their target audiences form anchors out of the information. This is crucial, given the crucial role that anchoring plays in shaping subsequent behavior [7, 15]. The rest of the paper is organized as follows: the introduction is followed by the literature review, the methodology, results, discussions, and suggestions for future research.

## 2. Literature review

### 2.1 The dual process theory

The dual process theory proposes that people process information through central and peripheral paths [16]. The central path is associated with information requiring more cognitive effort. This information processing path is deliberate, systematic, slower, and requires active attention [13]. The peripheral path is associated with heuristic processing [20]. Heuristic processing is relatively quick and requires less cognitive effort, relying on peripheral cues to process the information [21]. These paths are not mutually exclusive, and none can adequately explain information processing single-handedly [20]. The dual process theory proposes that individuals tend to process information if they have the motivation and the ability to process the information [16]. Multiple factors affect how people process messages via the two processing paths. These include features of the source, characteristics of the recipient, characteristics of the message, features of the communication channel, and characteristics of the context within which the message is received [22–24]. Various studies have applied this theory to examine individuals' processing of information [21–23]. This study proposes twelve factors associated with the anchoring of altruistic messages. These factors are grouped into those that trigger central processing and those that trigger peripheral processing.

### 2.2 Anchoring

Anchors are reference points against which the new information is judged or processed [18]. Anchoring is the process by which individuals form cognitive anchors [15]. Anchors are critical in information processing because they affect attention and the information receiver's perception of what is valuable and what is not. Information congruent with anchors is considered essential and valuable [10]. In the media and corporate world, communicators use anchors for their benefit. They aim to establish their communications as anchors against which audiences' attention toward future communications is judged [9]. Anchors are subject to change or replacement if individuals are exposed to more convincing information [25]. Because anchors are well established in the cognitive systems of human beings, changing them takes work and requires systematically crafted information [15]. Understanding how information can be prepared systematically to change or replace existing cognitive anchors.

Although extant research has examined the determinants of anchoring, they have not dwelled much on elucidating how anchors are formed based on dual process stimuli that appeal to emotion and cognition. For instance, a number of these studies have applied priming theory to examine the determinants of anchoring [6, 9, 18, 26]. These studies have largely examined links between numerical and semantic stimuli and anchoring. One study established that anchoring occurs if numeric and semantic stimuli are used together rather than separately [6]. Another study established that when individuals are presented with numerical information, they tend to anchor the information such that it remains in their memory systems even when it is retracted [9]. Numerical stimuli tend to be anchored more because they are concrete and specific, making it easier for individuals to retain them [18, 26]. Once the stimuli are anchored, they shape individuals' perceptions of subsequent stimuli [9].

Other studies have established that when messages are presented in a way that they appear relevant to the individual and the social context, they are anchored by the recipient. These studies have primarily applied the social representations theory to explain anchoring [12, 27, 28]. Information tends to be anchored in health communication if it emphasizes the individual's responsibility for protection and social and economic factors that increase vulnerability [27]. Another study also established that when COVID-19 was presented as a contagious enemy, it evoked fear, sadness, nervousness, and fright, and the negative emotions the representation of the virus evoked led to the individuals' retainment of the information about the pandemic [29].

Other studies suggest that information sources can determine whether the audience anchors the information. Some of these studies suggest that information from visual media tends to be retained and subsequently shapes audiences' perspectives about related issues [11]. The retainment occurs because visual stimuli are more powerful and have more attention-grabbing power than other stimuli forms due to their vividness [30]. Other studies also argued that when an individual directly witnesses an event firsthand, they remember the event [31]. When individuals notice altruistic actions, strong emotions such as admiration, gratitude, and empathy are evoked. Consequently, the witnessed event is mentally retained and becomes a reference point to which future behavior is anchored [31]. The current study applies the dual process theory to explore how central and peripheral factors lead to anchoring, particularly given that effective communication and information retainment tend to occur as a result of central and peripheral processing of stimuli [16, 23, 32]. These factors proposed by this study are discussed in the following section and shown in Table A1.1 of the S1 File.

## 3. Critical factors associated with anchoring

### 3.1 Naming

Naming is the process by which the subject of communication is given an identity. Naming gives contextual clarity, directs attention to important and relevant aspects of the message, and reduces ambiguity and misinterpretation of a message [33, 34]. Naming also serves as a mnemonic device. A memorable name for a message makes it easier for recipients to recall a message in subsequent situations. Where attention spans are limited, a name leads to the achievement of quick understanding [28, 35]. Thus, naming can significantly aid understanding and concretizing information into memory, leading to anchoring.

### 3.2 Emotional anchoring

Emotional anchoring is the communicative process by which a new idea is attached to well-known emotions [10]. Emotional anchoring is embedded in the verbal and visual illustrations used during communication to enhance communication effectiveness [10, 36]. People focus on a message when it triggers emotions [37]. Furthermore, when people are emotionally connected to a message, they relate it to their own experiences, enhancing their comprehension and retention of the message [12, 38]. Emotions also evoke empathy and encourage perspective taking on the communicated issue, improving engagement with the subject of communication [39]. Thus, emotional anchoring leads to the active processing of messages, ultimately leading to better understanding and retention of information.

### 3.3 Thematic anchoring

In some cases, the media crafts the message using familiar themes to aid the audience's comprehension of the message [28]. By connecting new information to familiar themes, the

cognitive effort required to process information is reduced, and consequently, the information is understood and retained easier [40]. In addition, using themes that are known to the audience enhances engagement and understanding of the message [28]. Thematic anchoring has been used to effectively communicate messages about climate change to bring about lasting changes in people's thinking and behavioral patterns [41].

### 3.4 Use of antinomies

Antinomies are contradictions between seemingly valid principles, ideas, or concepts [42]. The contradictory concepts presented by antinomies stimulate thought and create tension, which pushes individuals to think deeply about the issues being communicated to resolve the contradiction and arrive at a proper understanding of the information [43]. Thus, one dyad of the antimony directs thinking and attention to the other dyad [44]. Antinomies persuade recipients by guiding them toward a particular viewpoint that resolves the contradiction [12]. The engagement that antinomies stimulate could enhance understanding and retainment of the message, leading to anchoring. During the COVID-19 pandemic, media presentation of casualties has helped direct audiences' attention and efforts toward prevention during the pandemic [45].

### 3.5 Use of metaphors

Metaphors make a message comprehensible by letting the target audience imagine it as something else [42]. Metaphors make complex concepts more understandable to the audience and play a key role in bridging the gap between the familiar and the unfamiliar [46]. Metaphors relate complex ideas to familiar ideas, enhancing simplification and conciseness of information [47]. As such, they help audiences visualize abstract concepts, which sparks interest, curiosity, and attention [48]. Consequently, the audience's understanding and retention of the information is enhanced. During the COVID-19 pandemic, the media has used metaphors such as "it is a war" and "it is a battle" to direct people's attention to preventive behaviors [49].

### 3.6 Understanding the cognitive ability of the audience

People differ in their information processing abilities. Some may process intended messages quicker and correctly, whereas others may process them slower or fail to derive the intended meaning from the messages [50]. Understanding the intellectual abilities of the audience ensures that the language of the message matches the audience's level of comprehension and the message is framed so that it is relevant in the context within which it is delivered [51]. Ultimately, messages adapted to match the cognitive ability of the audience enhance comprehension and retention by the audience [45].

### 3.7 Repetition

Presenting the same message several times eases the cognitive load required to grasp a message in one go [32]. Apart from reinforcing a message's identity in a consumer's mind, repetition allows the audience to get clarification on misunderstood points, ultimately strengthening the message's trace in the individual's memory [52].

### 3.8 Crafting engaging information/messages

The active processing of information depends on how engaging the information is. [39]. Engaging information tends to attract the attention of the audience, and evokes the audience's emotions [23], and is processed with more cognitive effort [16]. Information processed with

more cognitive effort is more resistant to counter-persuasion and tends to be retained comparatively more than information processed passively [23]. Therefore, we propose crafting engaging information that can result in anchoring the information by its recipient.

### 3.9 Timing

In media communication, timing is mainly related to message scheduling, which refers to decisions about when and how long a program should run [53]. Timing plays an important role in communication because it impacts how messages are received and processed by the audience [54]. Delivering a message at the right time increases its relevance and the likelihood of capturing the audience's attention. In addition, proper timing can tap into other important determinants of communication effectiveness, such as the audience's emotional state and cognitive overload [13, 30]. Suppose a message matches the emotional state of the audience and does not overwhelm the audience with a lot of information. In that case, it can be processed more focused, resulting in proper comprehension and retention of the message by the audience.

### 3.10 Cultural consideration

Whether people allow an idea to form part of their cognitive schemas depends on whether it is in line with the cultural beliefs and social norms of the audience [55]. Messages are interpreted differently by people of different cultures [56]. Messages deemed culturally inappropriate by the audience are disregarded, whereas culturally appropriate messages are taken in and anchored by the audience [54]. The ease with which messages that align with the audience's culture are accepted implies that such messages are processed heuristically.

### 3.11 Consistency

Consistency implies that the same idea is presented, though sometimes in different ways [57]. As stated earlier, message repetition can be an essential determinant of anchor formation. However, if the core of a message changes every time it is repeated, the audience may not pick out a common theme from the message and anchor it [58]. Messages presented consistently are remembered much easier because subsequent messages do not conflict with previous messages. Not much cognitive effort is needed to resolve the conflict that subsequent messages present [59]. Therefore, it is expected that message consistency can result in anchoring.

### 3.12 Feedback

Feedback is an essential sub-process in any communication endeavor because it is a platform by which the audience can seek clarification on the communicated message [15] Feedback is essential for ensuring a notification is rooted in the receiver's mind [60]. By providing the necessary clarification, the sender ensures that the audience understands the message as intended. In addition, feedback allows the sender to gauge the degree to which the audience understands the message [61]. If, based on the feedback, the sender finds out that the message needs to be understood how it was intended, they can make necessary changes to ensure that the audience understands it well. By seeking clarification, the individual is cognitively engaged, and their understanding is concretized over time, which could result in anchoring.

## 4. Materials and methods

### 4.1 The DEMATEL method

The DEMATEL method was developed in the 1970s by the Banelle Institute of Geneva. One of the method's most unique functions is its ability to construct relational structures among

factors. The technique has been used in various research contexts, including consumer behavior, supply chain management, and health promotion [8, 62, 63]. DEMATEL is suitable for solving multiple-factor problems under uncertain situations. Communication is sometimes uncertain because many external factors affect the communication processes. Therefore, this study applies a modified DEMATEL—the F-DEMATEL method.

## 4.2 F-DEMATEL

Applying the fuzzy theory in DEMATEL helps analyze inaccurate data. The experts make the ratings based on their experiences rather than crisp values. Fuzzy techniques are incorporated into the DEMATEL method to reduce uncertainty and subjectivity and increase reliability. Various studies have demonstrated that the F-DEMATEL technique effectively aggregates experts' views into reliable information in several contexts, such as logistics and supply chain management [19, 64], construction [65], business management [66, 67], and health communication [8]. F-DEMATEL provides insights into how factors earmarked for decision making can be prioritized as with other multi-criteria decision methods. This study used the F-DEMATEL method because it can illustrate the causal relationships among the factors while overcoming ambiguity, subjectivity, and uncertainty, which are typical in many decision settings [68]. Whereas other variations of DEMATEL, such as Grey DEMATEL, could also be utilized, these approaches primarily focus on solving problems in systems with inadequate information or imprecise data [69–72]. This study assessed relationships among unclear criteria involving linguistic variables and qualitative data. Coupled with the need to minimize the subjectivity of experts in the rating process [19, 68], the researchers applied the F-DEMATEL method in this study. The results provide insights into which factors ought to be prioritized if altruistic messages are to be anchored by their recipients. This study used the F-DEMATEL calculation procedure in a prior study [73]. The computational steps used are explained below.

**4.2.1 Step 1: Determination of the influencing factors in the system.** Through a literature review, factors affecting the outcome variable are determined. Participants provide subjective rates of the degree of influence between every pair of factors.

**4.2.2 Step 2: Designing the fuzzy linguistic scale.** The degrees of influence in the fuzzy rating scale are in five semantic levels: no influence, very low influence, low influence, high influence, and very high influence [73]. The fuzzy linguistic scales and their corresponding values, as used in this study, are shown in Table A1.2 of the S1 File.

**4.2.3 Step 3: Computing the initial direct relation fuzzy matrix.** The experts evaluate the relationships between every pair of proposed factors based on their understanding of the semantic evaluation scale. The ratings for every pair of criteria by expert $k$ are denoted by $Z_{ij}^k$. For each expert, the initial direct-relation fuzzy matrix is as follows:

$$Z^K = \begin{bmatrix} 0 & Z_{12}^k & \cdots & Z_{1n}^k \\ Z_{21}^k & 0 & \cdots & Z_{2n}^k \\ \vdots & \vdots & \ddots & \vdots \\ Z_{n1}^k & Z_{n2}^k & \cdots & 0 \end{bmatrix} \text{k} = 1, 2, \ldots, \text{p} \ldots (1), \text{ where } Z_{ij}^k = \left( l_{ij}^k, m_{ij}^k, u_{ij}^k \right) \quad (1)$$

The combined average direct relation matrix for all respondents is obtained as follows:

$$\text{A} = \frac{1}{P} \sum_{k=1}^{P} Z^k \quad (2)$$

#### 4.2.4 Step 4: Normalizing the direct-relation fuzzy matrix.

$$r^k = \max_{1 \le i \le n} \left( \sum_{j=1}^{n} u_{ij}^k \right) k = 1, 2, \ldots, p \tag{3}$$

To compare the proposed criteria, linear scale transformation is used. Thereafter, the normalized direct relation fuzzy matrix $X^k$ is obtained by dividing each element of the average direct relation matrix by the highest value of the sum of the matrix's rows and columns ($r^k$):

$$X^K = \begin{bmatrix} X_{11}^K & X_{12}^k & \cdots & X_{1n}^k \\ X_{21}^k & X_{22}^k & \cdots & X_{2n}^k \\ \vdots & \vdots & \ddots & \vdots \\ X_{n1}^k & X_{n2}^k & \cdots & X_{nn}^k \end{bmatrix} \text{k} = 1, \ldots, \text{p} \ldots (4); X_{ij}^k = \left( L_{ij}^k, M_{ij}^k, U_{ij}^k \right) = \left( \frac{Z_{ij}^k}{r^k} \right)$$

$$= \left( \frac{l_{ij}^k}{r^k}, \frac{m_{ij}^k}{r^k}, \frac{u_{ij}^k}{r^k} \right) \tag{4}$$

#### 4.2.5 Step 5: Obtaining the total relation fuzzy matrix.
The total-relation fuzzy matrix $T$ is obtained by normalizing the direct-relation fuzzy matrix using Eqs (5) to (8).

$$TL = [TL_{ij}] = \lim_{c \to \infty} (L + L^2 + \cdots + L^c) = L(I - L)^{-1} \tag{5}$$

$$TM = [TM_{ij}] = \lim_{c \to \infty} (M + M^2 + \cdots + M^c) = M(I - M)^{-1} \tag{6}$$

$$TU = [TU_{ij}] = \lim_{c \to \infty} (U + U^2 + \cdots + U^c) = U(I - U)^{-1} \tag{7}$$

$$T^K = \begin{bmatrix} (TL_{11}, TM_{11}, TU_{11}) & (TL_{12}, TM_{12}, TU_{12}) & \cdots & (TL_{1n}, TM_{1n}, TU_{1n}) \\ (TL_{21}, TM_{21}, TU_{21}) & (TL_{22}, TM_{22}, TU_{22}) & \cdots & (TL_{2n}, TM_{2n}, TU_{2n}) \\ \vdots & \vdots & \ddots & \vdots \\ (TL_{n1}, TM_{n1}, TU_{n1}) & (TL_{n1}, TM_{n1}, TU_{n1}) & \cdots & (TL_{nn}, TM_{nn}, TU_{nn}) \end{bmatrix} \tag{8}$$

#### 4.2.6 Step 6: De-fuzzifying the total relation matrix.
To de-fuzzy the sums of the rows ($D_i$) and columns ($R_i$), the best non-fuzzy performance (BNP) method is used. The BNP values for the triangular fuzzy numbers are calculated using Eq (9). Eq 10 is used to obtain the defuzzification values of total-relation matrices.

$$BNP = 1 + \frac{(u - 1) + (m - 1)}{3} \tag{9}$$

$$T^K = \begin{bmatrix} T_{11}' & T_{12}' & \cdots & T_{1n}' \\ T_{21}' & T_{22}' & \cdots & T_{2n}' \\ \vdots & \vdots & \ddots & \vdots \\ T_{n1}' & T_{n2}' & \cdots & T_{nn}' \end{bmatrix} \text{where } T_{ij}' = TL_{ij} + \frac{\left( TU_{ij} - TL_{ij} \right) - \left( TM_{ij} - TL_{ij} \right)}{3} \tag{10}$$

**4.2.7 Step 7: Establishing and analyzing the F-DEMATEL diagram.** The summations of rows and columns are plotted as vectors $D_i$ and $R_i$ (Eqs 11 and 12). Prominence ($D_i + R_i$), the vector for the horizontal axis is obtained by summing the rows and columns for each factor. Relation ($D_i - R_i$), the vector for the vertical axis, is obtained by subtracting columns from rows. Thereafter, the criteria are classified into cause and effect sets. Factors with positive $D_i - R_i$ values are categorized as causal factors, and factors with negative $D_i - R_i$ effect factors. The causal model is obtained by graphing the values of $D_i + R_i$ and $D_i - R_i$. The causal relationships among the factors can be obtained by comparing the elements of the de-fuzzified total relation matrix to a threshold value, which is obtained by summing the mean and standard deviation of the de-fuzzified matrix.

$$D_i = \sum_{x=1}^{n} T'_{ix} \ldots (11); \; R_i = \sum_{y=1}^{n} T'_{yj} \tag{11}$$

$$\propto = T_{\mu}^{k} + T_{\sigma}^{k} \tag{12}$$

## 4.3 Procedure

The researcher proposed twelve determinants of anchoring altruistic messages based on a literature review. The researchers reviewed literature on anchoring published between 2018 and 2023 in the Google Scholar database. A number of the key phrases were applied to search for the factors. These included (but were not limited to) "determinants of anchoring in communication", "factors leading to anchoring in communication," and "antecedents of anchoring in communication". The twelve factors discussed in Section 3 were identified. To ascertain the usability of the factors in the F-DEMATEL procedure, a modified Delphi (M-Delphi) process was conducted with a panel of ten experts. The ten experts working with Da Ai Television in Taiwan were asked to rate the extent to which they believed each factor would lead to the anchoring of altruistic messages. The experts were selected by convenience sampling The rating was done on a 7-point Likert scale. After the first round, the experts rated each factor's causal effect on anchoring higher than the 75% threshold (i.e., each factor's mean score was higher than 5.250) [74]. The procedure was repeated over five rounds, and after each of these rounds, the experts' ratings of each factor's causal effect exceeded the 75% threshold. Thus, the experts reached a consensus that the factors had causal effects on anchoring. Consequently, the factors were used in the F-DEMATEL study. Fig 1 shows the framework for this research. The study was designed in three major steps: identifying the factors, analyzing the barriers using F-DEMATEL, and discussing the study's implications.

Because the items were translated from English to Chinese, a pilot study (n = 4) was conducted to ascertain the suitability of the translated items. The participants were employees of Da Ai Television, a media arm of Tzu Chi Cultural and Communication Foundation, a Taiwan-based charity organization. Da Ai Television is the media arm of the foundation, which spreads information about the humanitarian work that the organization does. To qualify for participation in the study, the participants were required to have over ten years of managerial experience producing TV programs about altruism with the station. Because all participants were from one media station, convenience sampling was used, as the study primarily targeted experts in the field. Before participation, the participants were informed about the aims of the study and oriented with the questionnaire filling procedures. This study collected data from 14 experts affiliated with Da Ai Television for at least 20 years, bringing extensive communication experience. The participants were handed a rating sheet in which they were asked to rate the extent to which they believed one factor affected any other factor using the semantic scale in Table 1.

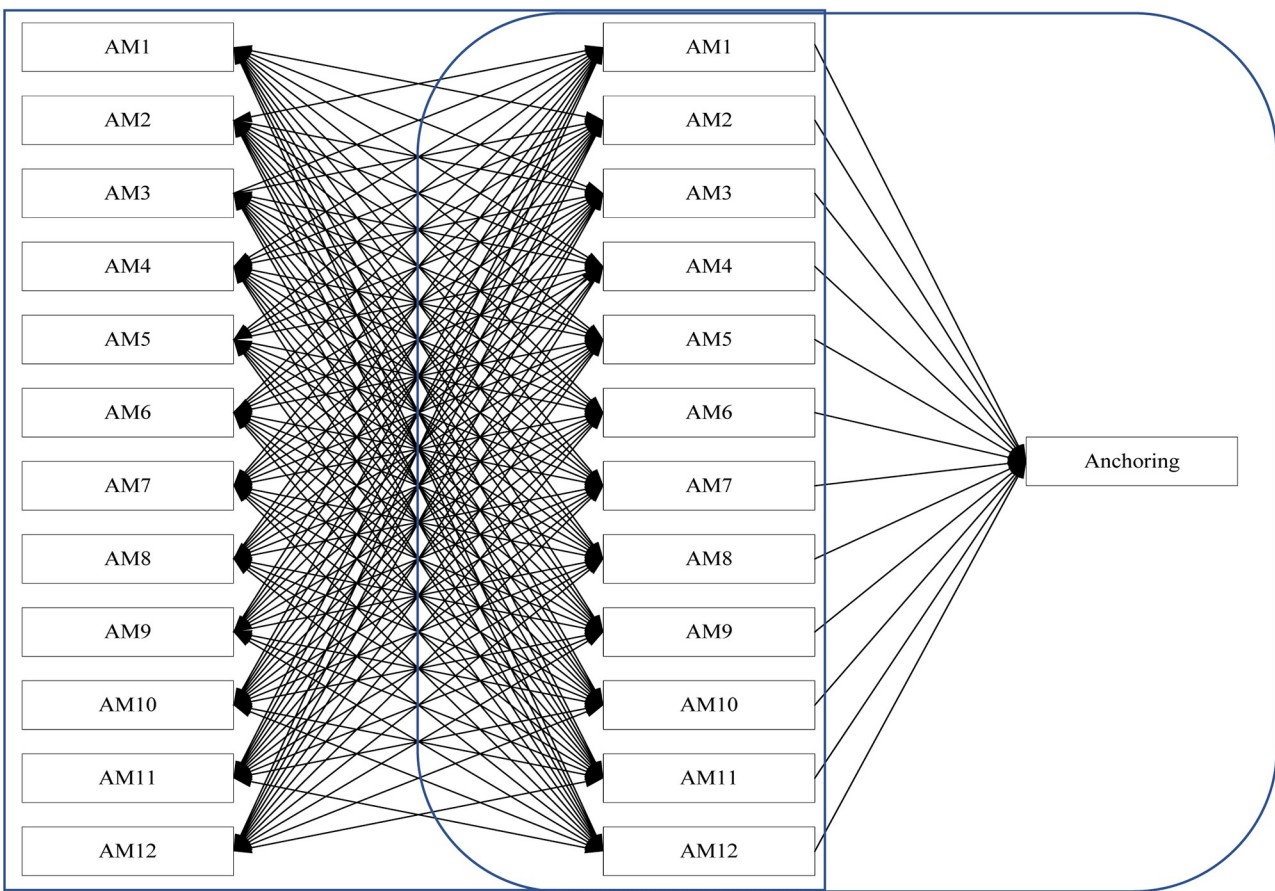

**Fig 1. DEMATEL process framework.**

## 5. Results

The F-DEMATEL method analyzed the proposed factors associated with anchor formation from altruistic messages. The computation procedure involved in the study is indicated step-by-step in the S1 File. Table 2 below indicates the causal relationships that were derived from the calculation.

**Table 1. Demographics.**

| Measures | Groups | Frequency | Percentage (%) | Cumulative % |
|---|---|---|---|---|
| Gender | Male | 6 | 42.9 | 42.9 |
| | Female | 8 | 57.1 | 100.0 |
| Age | 35–40 | 2 | 14.3 | 14.3 |
| | 41–55 | 5 | 35.7 | 50.0 |
| | 56 or more | 7 | 50.0 | 100.0 |
| Education | Bachelor's degree | 7 | 50.0 | 50.0 |
| | Postgraduate | 7 | 50.0 | 100.0 |
| Experience | 10–15 years | 5 | 35.7 | 35.7 |
| | 15–20 years | 5 | 35.7 | 71.4 |
| | 20 years or more | 4 | 28.6 | 100.0 |

**Table 2. Prominence and cause and effect relationships.**

| Factor | | $D_i$ | $R_i$ | $D_i+R_i$ | $D_i-R_i$ | Factor category |
|---|---|---|---|---|---|---|
| Naming | A1 | 0.230 | 0.314 | 0.544 | **-0.085** | **Effect** |
| Emotional anchoring | A2 | 0.349 | 0.261 | 0.610 | **0.088** | **Causal** |
| Thematic anchoring | A3 | 0.272 | 0.274 | 0.546 | -0.002 | Effect |
| Use of antinomies | A4 | 0.231 | 0.290 | 0.520 | **-0.059** | **Effect** |
| Use of metaphors | A5 | 0.292 | 0.296 | 0.588 | -0.005 | Effect |
| Understanding the cognitive ability of the audience | A6 | 0.299 | 0.277 | 0.576 | 0.022 | Causal |
| Repetition | A7 | 0.304 | 0.272 | 0.576 | **0.033** | **Causal** |
| Crafting engaging information/messages | A8 | 0.274 | 0.302 | 0.577 | -0.028 | Effect |
| Timing | A9 | 0.244 | 0.275 | 0.519 | **-0.031** | **Effect** |
| Cultural consideration | A10 | 0.346 | 0.257 | 0.604 | **0.089** | **Causal** |
| Consistency | A11 | 0.348 | 0.256 | 0.605 | **0.092** | **Causal** |
| Feedback | A12 | 0.210 | 0.323 | 0.533 | **-0.113** | **Effect** |

*Note*: Bold elements are causal/effect factors above the threshold.

The results are organized in two ways. First, they are organized based on their degree of importance based on $D_i + R_i$ values. These values indicate that emotional anchoring (AM2) has the highest degree of importance among all the factors. The rest of the factors are arranged as follows: A10 > A11 > A5 > A7 > A6 > A8 > A3 > A1 > A12 > A4 > A9. Using their $D_i - R_i$ values, they were categorized as causal ($D_i - R_i > 0$) or effect factors. The degree of significance of the causal and effect factors was decided using a cut-off point determined by the sum of the average and standard deviation of the de-fuzzified total relation matrix ($\alpha = 0.031$) of all the elements of the fuzzy total relation matrix. A positive value indicates that the factor affects other factors. A negative value indicates that the factor is an effect factor, i.e., influenced by other factors. The higher the value of the causal factor, the greater its degree of effect. The higher the absolute value of the effect factor, the greater the extent to which it is affected by other factors. In order of their size, causal factors are AM11, AM10, AM2, AM6, AM7. AM11, AM10, AM8 and AM2 are the only significant causal factors, The effect factors are AM12, AM1, AM4, AM9, AM8, AM5, AM3, with AM12, AM9. AM1 and AM4 are the significant effect factors. Fig 2 depicts the factor scatter diagram based on the results.

The results indicate that emotional anchoring had a causal effect on nine factors: naming, thematic anchoring, use of antinomies, use of metaphors, understanding the cognitive ability of the audience, repetition, crafting engaging information, consistency, and feedback. Repetition had causal effects on crafting engaging information, timing, and feedback. On the other hand, cultural consideration had causal effects on naming, emotional anchoring, thematic anchoring, use of antinomies, repetition, and feedback. Consistency had causal effects on naming, thematic anchoring, use of antinomies, use of metaphors, understanding the cognitive ability of the audience, crafting engaging information, and feedback. Fig 3 shows these causal relationships.

## 6. Discussion

Based on the results, consistency is the most critical factor affecting the formation of cognitive anchors from altruistic messages. This finding is consistent with the findings of prior studies, which established that consistency is essential to enhancing the retainment of information [75–77]. The core message communicated via television must be the same as it is re-broadcast over time. If a message contradicts a prior communication on the same topic, the audience

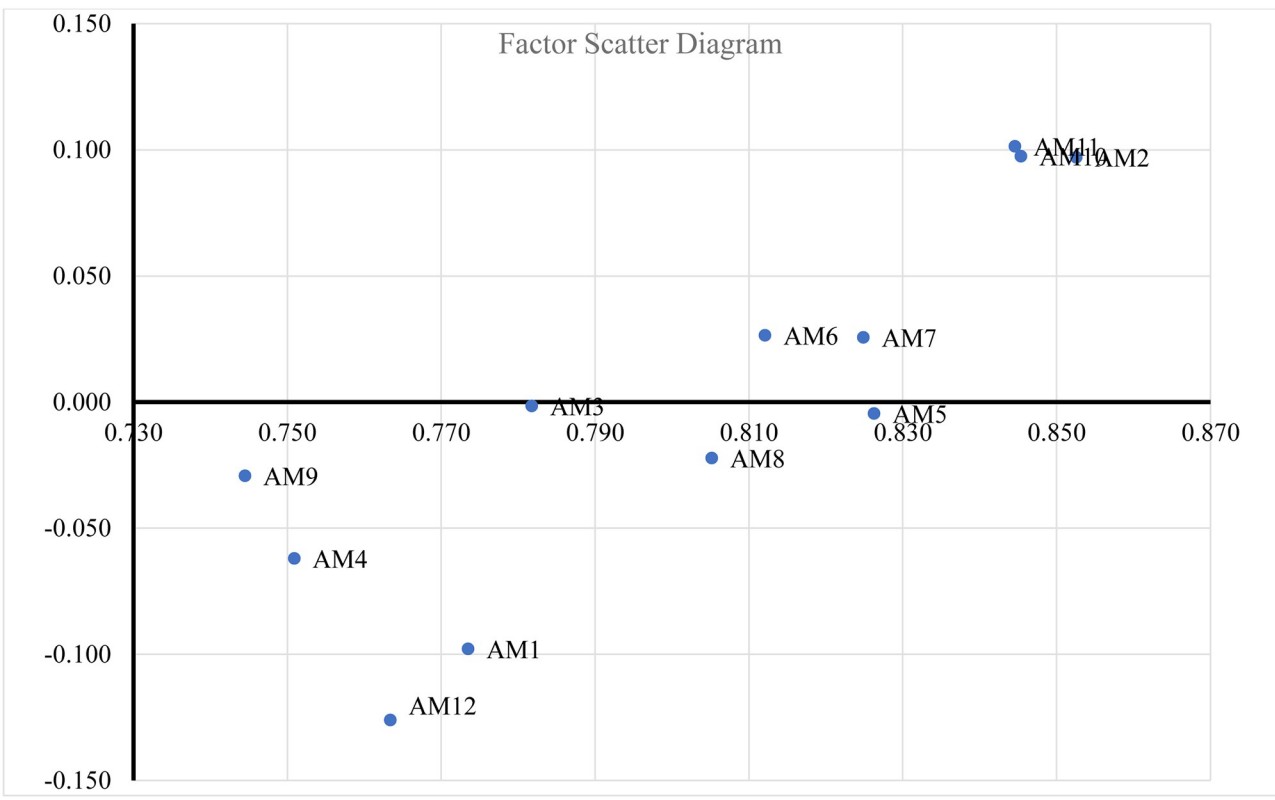

**Fig 2. Factor scatter diagram.**

may question the organization's motives, resulting in negative evaluations of the message and the organization [76]. An audience with a negative attitude toward an organization is not likely to be receptive nor to retain the organization's messages [77]. Furthermore, consistency ensures that naming, thematic anchoring, use of antinomies, use of metaphors, alignment with the audience's cognitive ability, and the engaging nature of messages and feedback are achieved. This is also consistent with prior research, which suggests that consistency of social communication ensures that the message is accorded an identity by the audience, is processed accordingly by the audience, and triggers audience engagement [39, 75]. Secondly, consistent with prior research [54, 78, 79], this study found that cultural consideration is another important determinant of the formation of anchors from altruistic communication. Based on the results, cultural consideration also ensures that naming, emotional anchoring, thematic anchoring, use of antinomies, repetition, and feedback are enhanced. People's cultural beliefs influence whether they accept information from the media [78]. Content that contradicts the audience's cultural beliefs is disregarded [54] because it goes against the audience's longstanding way of doing things and may be deemed disrespectful and harmful to their cultural identity [79].

The study's findings also demonstrate that emotional anchoring leads to anchoring. It also addresses the need for naming, thematic anchoring, use of antinomies, use of metaphors, understanding the audience's cognitive ability, repetition, crafting engaging information, consistency, and feedback. This provides further evidence to prior research findings that emotional appeal enhances the effectiveness of social communication [11, 30, 36]. Messages that

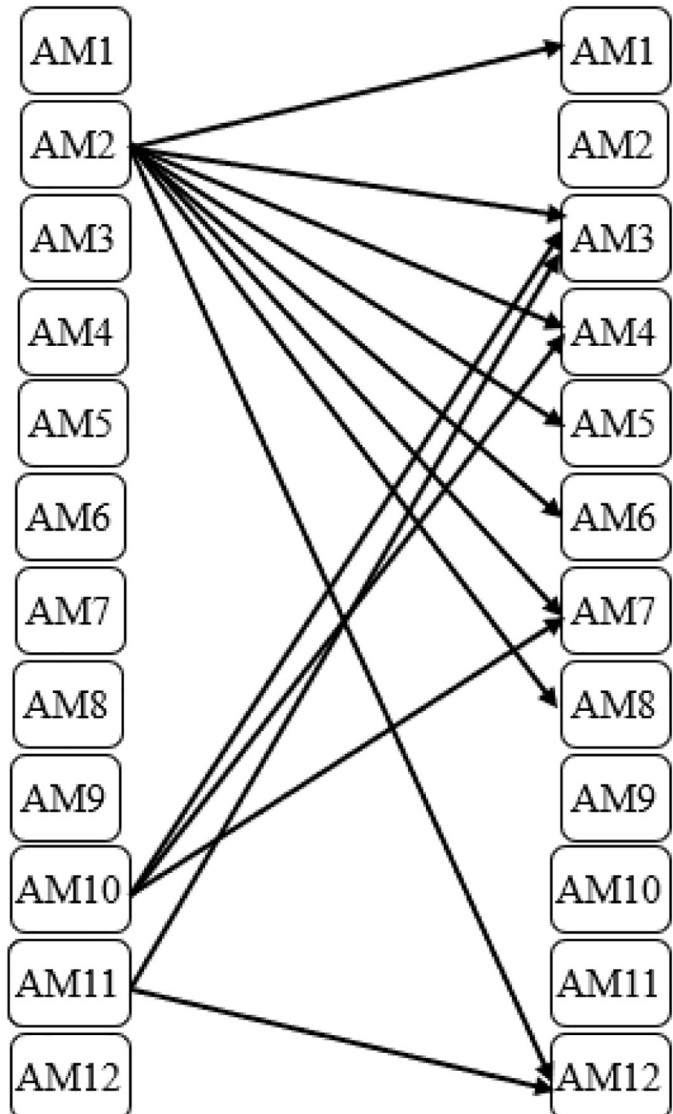

**Fig 3. Cause and effect relationship diagram.**

are emotionally loaded evoke empathy and consequently help recipients connect on a deeper level with the message, enhancing engagement, understanding, and retainment of the message [23, 78].

The results also indicate that understanding the cognitive ability of the audience and repetition are not highly significant causal determinants of anchoring. This result is different from the findings of prior research, which established that repetition and ensuring congruence between a message and the capacity of the audience to understand it are critical in ensuring the effectiveness of communication [32, 51, 52]. Television audiences are very diverse, and it takes work to understand the audience's general or specific cognitive ability. This is because cognitive ability significantly varies from individual to individual [80] compared to other socially shared values, such as cultural beliefs. As such, considering the audience's cognitive ability may not be the most important predictor of successful anchoring. In addition,

repetition is not a significant causal determinant of anchoring. This finding indicates that there may be more effective means of achieving anchoring, such as the 'quality of the message,' i.e., its consistency with prior communications in similar programs, its cultural appropriateness, and its ability to evoke the audience's emotions, are more important considerations.

Our findings demonstrate that feedback is the most influential effect factor, with a relatively moderate degree of importance. This indicates that the feedback issue is addressed by paying attention to the causal factors. In the communication process, feedback helps clarify ambiguous or misunderstood elements of the message to help the audience understand the message as intended by the sender [61]. Consequently, the need for feedback to enhance the message's clarity is reduced. The findings also demonstrate that naming and using antinomies are the second and third most influential effect factors, with moderate influence. Naming gives the message an identity, ensuring it is appropriately imbued into the recipients' memory. However, naming alone cannot provide a message distinctiveness. Our findings suggest that the causal factors, i.e., consistency, cultural consideration, and emotional anchoring, could help ensure that the message is given a distinctive identity, which provides that it is remembered by the receiver [81]. On the other hand, antinomies help individuals to understand a message via polarities [42]. Antinomies are an element of social representations, which comprise values, beliefs, and practices that establish social order and direct communication among community members [82]. Social representations, of which antinomies are part, have cultural underpinnings and vary depending on context [83]. Because antinomies are linked with culture, considering the audience's culture can ensure the audience's formation of relevant antinomies, bearing in mind that culture is a significant causal factor.

## 7. Theoretical and practical implications

This study makes two key contributions to social communication literature. First, it applies the dual process theory to explain how central and peripheral factors lead to anchoring in altruistic communication. The study demonstrates that consistency, cultural consideration, and emotional anchoring have significant causal effects on anchoring. In so doing, we demonstrate the applicability of the theory in social communication. Secondly, by applying the F-DEMATEL method, the study shows the complex interrelations among the factors in the study. The findings indicate that consistency, cultural consideration, and emotional anchoring have causal effects on anchoring. In addition, consistency has causal effects on naming, thematic anchoring, use of antinomies, use of metaphors, understanding the cognitive ability of the audience, and crafting engaging messages and feedback. In addition, cultural consideration also has causal effects on naming, emotional anchoring, thematic anchoring, use of antinomies, repetition, and feedback. On the other hand, emotional anchoring leads to naming, thematic anchoring, use of antinomies, use of metaphors, understanding the audience's cognitive ability, repetition, crafting engaging information, consistency, and feedback.

This study's findings have considerable managerial implications. First, our findings indicate that the executives must consider message consistency, cultural appropriateness, and emotional anchoring. Although different channels and means may be used to send altruistic messages, ensuring the core message is the same over time and across the programs broadcast is crucial. In addition, the message must be culturally appropriate for the audience it is sent to. Culture is one of the lenses against which people evaluate the appropriateness of a message, such that messages that do not contradict the cultural beliefs of the audience are not regarded as inappropriate. Furthermore, given that altruism has emotional underpinnings as it partly emanates from one's concern for others, it is essential to ensure that the messages that promote it appeal to the emotions of the recipients. Considering these three factors when sending

messages about altruism can help ensure the effectiveness of the communication process because, ultimately, the audience turns the messages into cognitive anchors. In addition, addressing these three factors also helps achieve other important determinants of anchor formation. Secondly, our studies demonstrate the need for non-profit organizations and visual media houses not to focus exclusively on one aspect of the communication process. For a message to be anchored by the audience, it must have favorable features and consider the context in which it is being sent. Apart from being emotionally loaded and consistent with prior related communications, the message must be consistent with the audience's culture. Taking such a multi-dimensional approach to crafting a message can help ensure it is more effectively delivered and that the audience forms cognitive anchors out of the communicated message.

The findings of this study highlight the significant managerial implications for organizations like the Tzu Chi Cultural and Communication Foundation. First and foremost, the study emphasizes the importance of message consistency, cultural appropriateness, and emotional resonance when conveying altruistic messages. This is relevant to organizations like Tzu Chi, which frequently engage in humanitarian and altruistic efforts. Such organizations must ensure their core message remains consistent across various programs and channels. Additionally, tailoring messages to be culturally appropriate for the target audience is essential, as culture plays a pivotal role in shaping individuals' perceptions of message appropriateness. Furthermore, recognizing that altruism is deeply rooted in emotions, messages promoting altruistic actions should be designed to evoke the appropriate emotional responses from recipients. By considering these factors, organizations like Tzu Chi can enhance the effectiveness of their communication efforts, as the audience forms cognitive anchors based on the messages received.

Moreover, the study underscores the need for non-profit organizations, such as the Tzu Chi Cultural and Communication Foundation, and visual media houses to adopt a multi-dimensional approach to their communication processes. Merely focusing on one aspect of communication is insufficient for message anchoring. To resonate with the audience and form cognitive anchors, messages must be emotionally engaging, consistent with prior communications, and align with the audience's cultural context. This holistic approach to message crafting can significantly improve the effectiveness of message delivery and ensure that the audience internalizes the communicated message, a vital aspect for organizations like the Tzu Chi Cultural and Communication Foundation dedicated to philanthropic and humanitarian causes.

## 8. Limitations and suggestions for future research

Although this study provides important insights into the effective communication of altruism, it has some limitations that need to be addressed by future research. First, the data utilized by the study were collected at one point in time and from respondents based in Taiwan. However, communication is a very dynamic and cultural-dependent phenomenon. Its dynamics vary over time and depend on the place. Thus, future studies could examine this topic longitudinally to establish how the factors affecting the anchoring of altruistic messages vary over time. In addition, future studies could also investigate the topic in other cultural contexts to develop whether the factors may have different effects on the formation of anchors depending on context. Fourthly, the data used in the study were collected from experts who had experience working in the television industry and understood what it takes to communicate social issues effectively. However, television is one of many ways non-profit organizations send out their messages. Future studies could use data from other communication contexts, such as social media, given its increasing popularity in modern-day communication. In addition, future research could collect data from audiences that receive the messages rather than the experts.

This is particularly important, given that individuals in target audiences are the ones who receive the messages and who form cognitive anchors out of the messages. Future studies could also use other research designs, such as surveys, to examine the causal relationships between the proposed factors and the anchoring of altruistic messages. The replication of the model using other methods could ensure the accuracy and robustness of the proposed model and uncover other mechanisms (such as mediation) through which the factors affect anchoring.

## 9. Conclusion

Implementing effective social communication strategies is a major challenge for non-profit organizations, especially given the increasingly dynamic environment within which the non-profit organizations operate. As one of the indicators of successful social communication, anchoring altruistic communication is one of the communication outcomes that non-profit organizations should aim to achieve. This necessitates understanding the determinants of anchoring altruistic messages. Understanding the determinants of anchoring will lead to effective communication by non-profit organizations and ensure that individuals execute helping behaviors to others in need.

This study proposed twelve determinants of the anchoring of altruistic messages, namely emotional anchoring, thematic anchoring, use of antinomies, use of metaphors, understanding the cognitive ability of the audience, repetition, crafting engaging information, timing, cultural consideration, consistency, feedback, and emotional anchoring. The main causal factors established in this study are consistency, cultural consideration, and emotional anchoring, and the main effect factors established by the study are naming, use of antinomies, and timing. The findings of the study indicate that emotional anchoring, repetition, cultural consideration, and consistency are significant causal factors. Consistency is the most critical causal factor, with causal effects on naming, thematic anchoring, use of antinomies, use of metaphors, understanding the cognitive ability of the audience, crafting engaging information, and feedback. On the other hand, repetition had causal effects on preparing engaging information, timing, and feedback. Furthermore, cultural consideration had causal effects on naming, emotional anchoring, thematic anchoring, antinomies, repetition, and feedback. In contrast, emotional anchoring had causal effects on naming, thematic anchoring, use of antinomies, use of metaphors, understanding the cognitive ability of the audience, repetition, crafting engaging information, consistency, and feedback.

This study is the initial attempt to examine the determinants of anchoring of altruistic messages using the F-DEMATEL method through the lens of the dual-process theory. It not only brings to light the causal effect of each factor on anchoring but also indicates the complex interrelationships among the factors. Ultimately, the study demonstrates the factors that communication practitioners, particularly in non-profit organizations, ought to prioritize in altruistic communication. Despite these important insights, the study has several limitations, including the cross-sectional and contextual limitations that future studies could address.

## Supporting information

**S1 File.**
(DOCX)

## Author Contributions

**Conceptualization:** Chi-Horng Liao.

**Data curation:** Chi-Horng Liao, Chu-Chia Hsu.

**Formal analysis:** Chi-Horng Liao.

**Funding acquisition:** Chi-Horng Liao.

**Investigation:** Chi-Horng Liao, Chu-Chia Hsu.

**Methodology:** Chi-Horng Liao.

**Project administration:** Chi-Horng Liao, Chu-Chia Hsu.

**Resources:** Chi-Horng Liao, Chu-Chia Hsu.

**Software:** Chi-Horng Liao.

**Supervision:** Chi-Horng Liao.

**Validation:** Chi-Horng Liao, Chu-Chia Hsu.

**Visualization:** Chi-Horng Liao, Chu-Chia Hsu.

**Writing – original draft:** Chi-Horng Liao.

**Writing – review & editing:** Chi-Horng Liao.

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
