## [Decision Letter · Decision Letter 0]

7 Aug 2023

PONE-D-23-17361Exploring Determinants of Formation of Cognitive Anchors from Altruistic Messages: A Fuzzy DEMATEL ApproachPLOS ONE

Dear Dr. Liao,

Thank you for submitting your manuscript to PLOS ONE. After careful consideration, we feel that it has merit but does not fully meet PLOS ONE’s publication criteria as it currently stands. Therefore, we invite you to submit a revised version of the manuscript that addresses the points raised during the review process.

I recommend that it should be revised taking into account the changes requested by the reviewers. Since the requested changes include valuable and constructive reviews, I would like to give you a chance to revise your manuscript. The revised manuscript will undergo the next round of review by same reviewers.

We look forward to receiving your revised manuscript.

Kind regards,

Baogui Xin, Ph.D.

Academic Editor

PLOS ONE

“The authors would like to thank the Tzu Chi Cultural and Communication Foundation for the funding support.”

“Chi-Horng Liao received funding from Tzu Chi Cultural and Communication Foundation.

 http://www.tzuchiculture.org.tw/

Reviewers' comments:

Reviewer's Responses to Questions

**Comments to the Author**

1. Is the manuscript technically sound, and do the data support the conclusions?

Reviewer #1: Yes

Reviewer #2: Yes

2. Has the statistical analysis been performed appropriately and rigorously? 

Reviewer #1: No

Reviewer #2: N/A

3. Have the authors made all data underlying the findings in their manuscript fully available?

Reviewer #1: No

Reviewer #2: Yes

4. Is the manuscript presented in an intelligible fashion and written in standard English?

Reviewer #1: Yes

Reviewer #2: No

5. Review Comments to the Author

Reviewer #1: Overall, I think this article is innovative, but as far as the writing of the article is concerned, the language is not concise enough and there is a lot of room for improvement, and I think the article needs to be revised in some way.

1. The article does not give the fuzzy language scale and its corresponding values.

2. This article is not very strict in the selection of factors, and although the article is a literature review for factor extraction, the factors still need to be screened to some extent so as to ensure the accuracy of the analysis results. For example, analysis of variance (ANOVA), etc.

3.The description of the factors is not sufficiently concise and the language should be more streamlined and a table format is recommended.

4. The article lacks a description of the existing research gaps as well as the significance of this study. It is suggested that a description of the current research gaps as well as the significance of the study be added to the introduction.

5. The article lacks a well-structured technology roadmap. A technology roadmap would give the reader a clearer understanding of the overall structure of the article.

6. The article is long in the conclusion section, and I think it could be more concise.

7. In the results section, I don't think it is necessary to repeat the formulae, thus leading to an increase in the length of the article; a good article should be written in simple language.

8. The literature review section of the article is too long; an article's literature review should be developed in terms of the current research status, research gaps, and the significance of this study. In addition, I think chapter 2.5 should be treated as a separate chapter, which would make the structure of the article clearer.

Reviewer #2: 1. The writing standard of the entire manuscript is not satisfactory. It needs a major overhaul and professional English proofreading.

2. The Abstract is not satisfactory. The background is not clear. The research gap is not mentioned. The result is not mentioned. The writing standard is not satisfactory

3. Citations in the manuscript are not uniform. You have used the number format in most cases but there are some citations in different formats like page 6 (De Leeuw et al., 2022); Page 9 (Fang, 2021). Moreover, page numbers are given in both the top right and bottom center.

4. In Introduction, Research Objectives are missing. Moreover, the research gap you are trying to address is not clear. The research contribution is not elaborately written.

5. Background of the study is not adequately explained in the Introduction.

6. More recent papers published in 2022 and 2023 in reputed journals like Elsevier or Plos should be discussed and cited. Especially, since it’s a MCDM paper submitted in Plos, cite other recently published Plos paper that has used various MCDM methods, especially the DEMATEL variants, to establish your paper's relevance to this journal. Try citing the following recent papers on F DEMATEL.

• Kuzu, A. C. (2023). Application of fuzzy DEMATEL approach in maritime transportation: A risk analysis of anchor loss. Ocean Engineering, 273, 113786.

• Çelik, M. T., & Arslankaya, S. (2023). Analysis of quality control criteria in an business with the fuzzy DEMATEL method: Glass business example. Journal of Engineering Research, 11(2), 100039.

• Siraj, M. T., Debnath, B., Kumar, A., Bari, A. M., Samadhiya, A., & Payel, S. B. (2023). Evaluating barriers to sustainable boiler operation in the apparel manufacturing industry: Implications for mitigating operational hazards in the emerging economies. Plos one, 18(4), e0284423.

7. Any specific systematic literature Review technique applied for factor identification? This is not mentioned or discussed anywhere. Moreover, what are the research protocols used (search strings, timeline of search, inclusive & exclusive criteria) for literature search?

8. The methodological study framework of the study is missing. Provide it in a pictorial format.

9. In section 3.2, you mentioned that “Figure 1 indicates the process framework for the m-delphi/F-DEMATEL techniques used in this study”. But I don’t find anything about Delphi here.

10. Correct all typos through out the paper, such as “delphi” (It’s a process name, should be “Delphi”). You have written both f and F DEMATEL. Which one is correct?

11. Why Fuzzy DEMATEL is used in this study over other methods? What are the salient advantages of F-DEMATEL over other MCDM methods?

12. In section 3.2.2, you mentioned that the fuzzy linguistic scale and its corresponding values are provided in Table 1. But Table 1 defines the Fuzzy total relation matrix. The linguistic scale is not found.

13. What do you mean by the process framework of DEMATEL in Figure 1? I have not seen such figure in any DEMATEL study. Delete this figure.

14. The experts' profile is not given. Provide it.

15. You have just written equations in methods. But their applications in your problem are not defined. You should mention the outcome of your results with regard to each of the equations.

16. Where is the reference of this F-DEMATEL approach? From which research article did you adopt the methodological approach? Since you didn’t invent F DEMATEL here, you have to cite the source paper here.

17. How did you collect the data? What sampling approach did you adopt for your research? There should be a discussion on the survey design.

18. In the Results, you have mentioned “After that, the direct relation matrix was normalized using Equation 3 (Table 1).” But you have defined Table 1 as Fuzzy total relation matrix. There are so many irregularities like this throughout the paper.

19. You have mentioned that Table 2 defines Cause and Effect relationships. But, you haven’t even defined which one are in cause group and which one are in effect group in the table. You should do that.

20. How you achieved the degree of significance as: (The degree of significance of the causal and effect factors was decided using a cut-off point determined by the average (α=0.033))? This is actually threshold value I think. You should mention how the threshold value is calculated and obtained.

21. Where is the causal relationship diagram? You have just drawn the clustered cause-effect group diagram in Figure 2. But the causal interrelations between the factors diagram is missing.

22. The description of the result and discussion is not satisfactory. You need to elaborate them.

23. The distinction of this study's findings with other relevant studies are missing at the end of the discussion section.

24. The theoretical implication needs to incorporate how it contributes theoretically to the existing literature.

25. Where is the main body of the conclusion??

26. The overall writing is very poor.

If authors must improve the paper addressing all these comments.

6. PLOS authors have the option to publish the peer review history of their article (what does this mean?). If published, this will include your full peer review and any attached files.

Reviewer #1: No

Reviewer #2: No

---

## [Author Response · Author response to Decision Letter 0]

18 Sep 2023

Dear Reviewer,

The time and effort devoted to reviewing the manuscript are greatly appreciated. The insightful comments and valuable suggestions have significantly enhanced the paper's quality. In response to this feedback, all raised points have been thoroughly addressed, and the revisions have been meticulously highlighted within the manuscript. Below, there is a detailed point-by-point response to these comments and concerns to aid in the review process.

Reviewer #1: Overall, I think this article is innovative, but as far as the writing of the article is concerned, the language is not concise enough and there is a lot of room for improvement, and I think the article needs to be revised in some way.

1. The article does not give the fuzzy language scale and its corresponding values.

Thank you for your suggestions. We have added a table showing the fuzzy language scale and its corresponding values (Table 1). 

2. This article is not very strict in the selection of factors, and although the article is a literature review for factor extraction, the factors still need to be screened to some extent so as to ensure the accuracy of the analysis results. For example, analysis of variance (ANOVA), etc.

Thank you for your comment. We conducted Delphi surveys with the participants before using the factors in the final survey. We initially did not include an explanation for the Delphi survey due to space limitations. The information about the Delphi survey has since been added to the first paragraph of section ‘3.3 Procedure’. We feel it would also be important to add another study, say a survey, to enhance the robustness of the findings of the F-DEMATEL study. However, due to time constraints, we could not conduct another study. This is indicated as one of the limitations of the current study in section 6. 

3. The description of the factors is not sufficiently concise and the language should be more streamlined and a table format is recommended.

Thank you for your recommendation. We enhanced the description of the factors to ensure that the link between every factor and anchoring is clear. We have also added a table with descriptions for each factor in the A1 file. 

4. The article lacks a description of the existing research gaps as well as the significance of this study. It is suggested that a description of the current research gaps as well as the significance of the study be added to the introduction.

Thank you for your suggestions. We added two research gaps in the third and fourth paragraphs of the introduction and added the theoretical and practical significance of the study in the final paragraph of the introduction.

5. The article lacks a well-structured technology roadmap. A technology roadmap would give the reader a clearer understanding of the overall structure of the article.

Thank you for your comment. The paper’s roadmap is in the final sentence of the final paragraph of the introduction section. 

6. The article is long in the conclusion section, and I think it could be more concise.

Thank you for the suggestions. We trimmed the discussion section and merged some subsections under it to make it more concise. 

7. In the results section, I don't think it is necessary to repeat the formulae, thus leading to an increase in the length of the article; a good article should be written in simple language.

Thank you for the comment. We deleted the detailed explanations of the results under the table, as these necessitated the use of the equations. They appeared to be repetitions of what was explained in the first paragraph, and so we deleted them.

8. The literature review section of the article is too long; an article's literature review should be developed in terms of the current research status, research gaps, and the significance of this study. In addition, I think chapter 2.5 should be treated as a separate chapter, which would make the structure of the article clearer.

Thank you for the suggestions. We shortened the literature review and revised it to ensure that it explains the current status of the research, the gaps, and the significance of this study. We turned section 2.5 to chapter 3. 

Dear Reviewer,

The time and effort devoted to reviewing the manuscript are greatly appreciated. The insightful comments and valuable suggestions have significantly enhanced the paper's quality. In response to this feedback, all raised points have been thoroughly addressed, and the revisions have been meticulously highlighted within the manuscript. Below, there is a detailed point-by-point response to these comments and concerns to aid in the review process.

1. The writing standard of the entire manuscript is not satisfactory. It needs a major overhaul and professional English proofreading.

Thank you for your comments. We have revised the manuscript substantially and had professional proofreading done on it. 

2. The Abstract is not satisfactory. The background is not clear. The research gap is not mentioned. The result is not mentioned. The writing standard is not satisfactory. 

Thank you for the observation. We have revised the abstract and added all relevant information in the abstract. 

3. Citations in the manuscript are not uniform. You have used the number format in most cases but there are some citations in different formats like page 6 (De Leeuw et al., 2022); Page 9 (Fang, 2021). Moreover, page numbers are given in both the top right and bottom center.

Thank you for the observation, we have revised all citations and page numbers to ensure that they are uniform. 

4. In Introduction, Research Objectives are missing. Moreover, the research gap you are trying to address is not clear. The research contribution is not elaborately written.

Thank you for the comments. We added the research gaps and objectives in the introduction. We also revised our writing of the research contributions in the introduction and section 6.2. 

5. Background of the study is not adequately explained in the Introduction.

Thank you for the observation. We revised the background to make it more comprehensible. 

6. More recent papers published in 2022 and 2023 in reputed journals like Elsevier or Plos should be discussed and cited. Especially, since it’s a MCDM paper submitted in Plos, cite other recently published Plos paper that has used various MCDM methods, especially the DEMATEL variants, to establish your paper's relevance to this journal. Try citing the following recent papers on F DEMATEL.

• Kuzu, A. C. (2023). Application of fuzzy DEMATEL approach in maritime transportation: A risk analysis of anchor loss. Ocean Engineering, 273, 113786.

• Çelik, M. T., & Arslankaya, S. (2023). Analysis of quality control criteria in an business with the fuzzy DEMATEL method: Glass business example. Journal of Engineering Research, 11(2), 100039.

• Siraj, M. T., Debnath, B., Kumar, A., Bari, A. M., Samadhiya, A., & Payel, S. B. (2023). Evaluating barriers to sustainable boiler operation in the apparel manufacturing industry: Implications for mitigating operational hazards in the emerging economies. Plos one, 18(4), e0284423.

Thank you for the suggestions. We have cited these papers and five other PlosOne papers. 

7. Any specific systematic literature Review technique applied for factor identification? This is not mentioned or discussed anywhere. Moreover, what are the research protocols used (search strings, timeline of search, inclusive & exclusive criteria) for literature search?

Thank you for the comment. We have elaborated the research protocol for the extraction of the factors in “Section 4.3 Procedure”. 

8. The methodological study framework of the study is missing. Provide it in a pictorial format.

Thank you for the comment. We have added the framework as Figure 1. 

9. In section 3.2, you mentioned that “Figure 1 indicates the process framework for the m-delphi/F-DEMATEL techniques used in this study”. But I don’t find anything about Delphi here.

Thank you for the suggestion. We have deleted the figure as per your recommendation in comment 13. We have discussed M-DELPHI as a preliminary technique for scrutinizing the factors prior to their usage in the study in Section 4.3. 

10. Correct all typos through out the paper, such as “delphi” (It’s a process name, should be “Delphi”). You have written both f and F DEMATEL. Which one is correct?

Thank you for the observation. This has been done. F-DEMATEL has been maintained throughout the paper. 

11. Why Fuzzy DEMATEL is used in this study over other methods? What are the salient advantages of F-DEMATEL over other MCDM methods?

Thank you for the questions. We have added the explanation for why we used F-DEMATEL in SECTION “4.2 DEMATEL”. 

12. In section 3.2.2, you mentioned that the fuzzy linguistic scale and its corresponding values are provided in Table 1. But Table 1 defines the Fuzzy total relation matrix. The linguistic scale is not found.

Thank you for the observation. We have added the Table under the section (which is now named Section 3.2.2). 

13. What do you mean by the process framework of DEMATEL in Figure 1? I have not seen such figure in any DEMATEL study. Delete this figure.

Thank you for the recommendation. We deleted the figure accordingly.

14. The experts' profile is not given. Provide it.

Thank you for the recomendation. We have added the respondents’ profile in Table 2 (Demographics). 

15. You have just written equations in methods. But their applications in your problem are not defined. You should mention the outcome of your results with regard to each of the equations.

Thank you for the commemnts. We added an appendix, A1, in which are contained the applications of the equations in each step of the F-DEMATEL calculation process. We have added the detailed explanation in the additional file to make the file easier to read and to avoid hitting the word limit in the main file. 

16. Where is the reference of this F-DEMATEL approach? From which research article did you adopt the methodological approach? Since you didn’t invent F DEMATEL here, you have to cite the source paper here.

Thank you for the observation. We have indicated the source of the F-DEMATEL procedure in the paper. We adopted it from the paper below: 

Liao, C. H. (2020). Evaluating the social marketing success criteria in health promotion: A F-DEMATEL approach. International Journal of Environmental Research and Public Health, 17(17), 6317. https://doi.org/10.3390/ijerph17176317.

17. How did you collect the data? What sampling approach did you adopt for your research? There should be a discussion on the survey design.

Thank you for the question. We have added the information in section 4.3. 

18. In the Results, you have mentioned “After that, the direct relation matrix was normalized using Equation 3 (Table 1).” But you have defined Table 1 as Fuzzy total relation matrix. There are so many irregularities like this throughout the paper.

Thank you for the observation. This has been rectified. We have used accurate terminologies for all tables. 

19. You have mentioned that Table 2 defines Cause and Effect relationships. But, you haven’t even defined which one are in cause group and which one are in effect group in the table. You should do that.

Thank you for the observation. This has been done accordingly. 

20. How you achieved the degree of significance as: (The degree of significance of the causal and effect factors was decided using a cut-off point determined by the average (α=0.033))? This is actually threshold value I think. You should mention how the threshold value is calculated and obtained.

Thank you for the observation. We have indicated the procedure for calculating the threshold value in the A1 file. We summed the mean and standard deviation of the de-fuzzified total relation matrix, a technique used by a prior study (Siraj et al., 2023).

21. Where is the causal relationship diagram? You have just drawn the clustered cause-effect group diagram in Figure 2. But the causal interrelations between the factors diagram is missing.

Thank you for your comment. We have added the causal relationship diagram as Figure 3. 

22. The description of the result and discussion is not satisfactory. You need to elaborate them.

Thank you for the comments. We enhanced the analysis of the results in the discussion section. 

23. The distinction of this study's findings with other relevant studies are missing at the end of the discussion section.

Thank you for the observation. We elaborated on the links between our findings and the findings of prior studies in the discussion section (Section 6).

24. The theoretical implication needs to incorporate how it contributes theoretically to the existing literature.

Thank you for the comment. We have enhanced the theoretical implications to ensure that there is clarity on how the paper contributes theoretically to existing literature. 

25. Where is the main body of the conclusion??

Thank you for the observation. We added a conclusion section (Section 9) to the manuscript accordingly. 

26. The overall writing is very poor.

If authors must improve the paper addressing all these comments.

Thank you for your comments and suggestions. We appreciate the time and effort you dedicated to providing us with valuable, concise feedback.

---

## [Decision Letter · Decision Letter 1]

10 Oct 2023

PONE-D-23-17361R1Exploring Determinants of Formation of Cognitive Anchors from Altruistic Messages: A Fuzzy DEMATEL ApproachPLOS ONE

Dear Dr. Liao,

Thank you for submitting your manuscript to PLOS ONE. After careful consideration, we feel that it has merit but does not fully meet PLOS ONE’s publication criteria as it currently stands. Therefore, we invite you to submit a revised version of the manuscript that addresses the points raised during the review process.

We recommend that it should be revised by taking into account the changes requested by Reviewers. I want to give you a chance to revise your manuscript. To speed the review process, the manuscript will only be reviewed by the Academic Editor in the next round.

We look forward to receiving your revised manuscript.

Kind regards,

Baogui Xin, Ph.D.

Academic Editor

PLOS ONE

Journal Requirements:

Reviewers' comments:

Reviewer's Responses to Questions

**Comments to the Author**

1. If the authors have adequately addressed your comments raised in a previous round of review and you feel that this manuscript is now acceptable for publication, you may indicate that here to bypass the “Comments to the Author” section, enter your conflict of interest statement in the “Confidential to Editor” section, and submit your "Accept" recommendation.

Reviewer #1: All comments have been addressed

Reviewer #2: (No Response)

2. Is the manuscript technically sound, and do the data support the conclusions?

Reviewer #1: Yes

Reviewer #2: (No Response)

3. Has the statistical analysis been performed appropriately and rigorously? 

Reviewer #1: Yes

Reviewer #2: (No Response)

4. Have the authors made all data underlying the findings in their manuscript fully available?

Reviewer #1: Yes

Reviewer #2: (No Response)

5. Is the manuscript presented in an intelligible fashion and written in standard English?

Reviewer #1: Yes

Reviewer #2: (No Response)

6. Review Comments to the Author

Reviewer #1: The author has answered the questions I asked very well and has made changes to the relevant parts of the article, and is recommended for acceptance!

Reviewer #2: The manuscript has improved significantly over its earlier version. However, I think the authors still need to address the following comments:

1) Authors should highlight all new changes using yellow highlights. It’s difficult to check the changes from the track change.

2) The abstract needs to be improved. More details on the study motivation and findings should be discussed there.

3) The authors need to discuss why they are choosing F-DEMATEL over other DEMATEL variants, such as Grey DEMATEL, citing recent (published in 2022-2023) relevant papers. Following recent articles might be helpful:

https://doi.org/10.1016/j.jclepro.2022.135791

https://doi.org/10.1016/j.health.2022.100120

https://doi.org/10.1016/j.oceaneng.2023.114805

https://doi.org/10.1016/j.smse.2023.100013

4) The conclusion needs to be improved and elaborated. Authors need to discuss crucial and interesting findings from their study and discuss some key contributions of the study.

5) I can still see some grammatical errors here and there. A thorough proofreading/language editing is necessary.

7. PLOS authors have the option to publish the peer review history of their article (what does this mean?). If published, this will include your full peer review and any attached files.

Reviewer #1: No

Reviewer #2: No

---

## [Author Response · Author response to Decision Letter 1]

18 Oct 2023

The author would like to express gratitude to the Academic Editor and the reviewers for their precious time and invaluable comments. All the comments have been carefully addressed. The corresponding changes and refinements made in the revised manuscript are summarized in the response below.

Reviewer #2: The manuscript has improved significantly over its earlier version. However, I think the authors still need to address the following comments:

1) Authors should highlight all new changes using yellow highlights. It’s difficult to check the changes from the track change.

Response: Thank you for the comment. All new changes are highlighted in yellow. 

2) The abstract needs to be improved. More details on the study motivation and findings should be discussed there.

Response: Thank you for the comment. Information about the study's rationale has been added, and more details about the findings are now included.

3) The authors need to discuss why they are choosing F-DEMATEL over other DEMATEL variants, such as Grey DEMATEL, citing recent (published in 2022-2023) relevant papers. Following recent articles might be helpful:

https://doi.org/10.1016/j.jclepro.2022.135791

https://doi.org/10.1016/j.health.2022.100120

https://doi.org/10.1016/j.oceaneng.2023.114805

https://doi.org/10.1016/j.smse.2023.100013

Response: Thank you for the comment. All the suggested article information has been added to justify the choice of F-DEMATEL over other DEMATEL variants. The suggested papers were particularly helpful in distinguishing F-DEMATEL from Grey DEMATEL. This input is greatly appreciated.

4) The conclusion needs to be improved and elaborated. Authors need to discuss crucial and interesting findings from their study and discuss some key contributions of the study.

Response: Thank you for the comment we revised the conclusion to ensure that it reflects the study’s important findings and contributions.

5) I can still see some grammatical errors here and there. A thorough proofreading/language editing is necessary.

Response: Thank you for the observation. The manuscript has been sent for proofreading to address all grammatical errors within it.

---

## [Editor Report · Decision Letter 2]

19 Oct 2023

Exploring Determinants of Formation of Cognitive Anchors from Altruistic Messages: A Fuzzy DEMATEL Approach

PONE-D-23-17361R2

Dear Dr. Liao,

We’re pleased to inform you that your manuscript has been judged scientifically suitable for publication and will be formally accepted for publication once it meets all outstanding technical requirements.

Kind regards,

Baogui Xin, Ph.D.

Academic Editor

PLOS ONE
---

## [Editor Report · Acceptance letter]

26 Oct 2023

PONE-D-23-17361R2 

Exploring Determinants of Formation of Cognitive Anchors from Altruistic Messages: A Fuzzy DEMATEL Approach 

Dear Dr. Liao:

I'm pleased to inform you that your manuscript has been deemed suitable for publication in PLOS ONE. Congratulations! Your manuscript is now with our production department. 

Kind regards, 

on behalf of

Professor Baogui Xin 

Academic Editor

PLOS ONE